# Head-Mounted Display-Based Augmented Reality for Image-Guided Media Delivery to the Heart: A Preliminary Investigation of Perceptual Accuracy

**DOI:** 10.3390/jimaging8020033

**Published:** 2022-01-30

**Authors:** Mitchell Doughty, Nilesh R. Ghugre

**Affiliations:** 1Department of Medical Biophysics, University of Toronto, Toronto, ON M5S 1A1, Canada; nilesh.ghugre@utoronto.ca; 2Schulich Heart Program, Sunnybrook Health Sciences Centre, Toronto, ON M4N 3M5, Canada; 3Physical Sciences Platform, Sunnybrook Research Institute, Toronto, ON M4N 3M5, Canada

**Keywords:** augmented reality, head-mounted displays, surgical navigation, magnetic resonance imaging, calibration

## Abstract

By aligning virtual augmentations with real objects, optical see-through head-mounted display (OST-HMD)-based augmented reality (AR) can enhance user-task performance. Our goal was to compare the perceptual accuracy of several visualization paradigms involving an adjacent monitor, or the Microsoft HoloLens 2 OST-HMD, in a targeted task, as well as to assess the feasibility of displaying imaging-derived virtual models aligned with the injured porcine heart. With 10 participants, we performed a user study to quantify and compare the accuracy, speed, and subjective workload of each paradigm in the completion of a point-and-trace task that simulated surgical targeting. To demonstrate the clinical potential of our system, we assessed its use for the visualization of magnetic resonance imaging (MRI)-based anatomical models, aligned with the surgically exposed heart in a motion-arrested open-chest porcine model. Using the HoloLens 2 with alignment of the ground truth target and our display calibration method, users were able to achieve submillimeter accuracy (0.98 mm) and required 1.42 min for calibration in the point-and-trace task. In the porcine study, we observed good spatial agreement between the MRI-models and target surgical site. The use of an OST-HMD led to improved perceptual accuracy and task-completion times in a simulated targeting task.

## 1. Introduction

Heart disease is the leading cause of global mortality and a significant precursor to heart failure (HF) [1]. Heart failure arises because of damage to the heart muscle and replacement with non-contractile scar tissue, predominantly caused by coronary artery disease and myocardial infarction (MI) [2]. Heart transplant remains the primary clinically available curative treatment for HF; however, available donor organs lag the rate of HF patient growth [3].

Myosin activator drugs [4], advanced biomaterials [5], and cell therapy-based approaches [6] are being explored for improving cardiac function after MI. In addition to the optimization of the proposed therapy and dose, these approaches require the accurate placement, distribution, and retention of delivered media across myocardial scar and border zone tissue [7]. Due to the diffusion characteristics of scar, targeted injections, which are spaced ~1 cm apart, have been shown to maximize the functional potential of the delivered therapy [8]. The trans-epicardial delivery pathway, where media-injections are made directly into scars on the epicardial surface of the beating or arrested heart via open chest surgery (Figure 1a), provides the highest media retention rate and potential for positive effect of current delivery strategies [9].

Due to its excellent tissue contrast, MRI has become a leading imaging modality for the non-invasive assessment of myocardial viability and cardiac function post-MI [10]. Segmented structures from MRI data can provide a detailed three-dimensional (3D) map of myocardial scar tissue (Figure 1b) from which a two-dimensional (2D) media delivery plan can be created (Figure 1c) to inform the optimal media delivery to regions of myocardial scar tissue via pre-planned configurations [8]. However, to leverage these 2D media delivery plans and scar maps, for guidance during targeted media injections, the surgeon is required to manipulate the orientation of imaging-derived models and to look away from the surgical scene to an adjacent monitor for guidance, limiting their effectiveness [11]. Further, due to the lack of clear delineation of scar tissue on the surface of the heart via direct eye inspection (Figure 1a), there is a reliance on the surgeon to create a mental mapping to adapt the scar map and 2D media delivery plan to the current surgical context (Figure 1b,c).

### 1.1. Motivation

We propose the use of augmented reality (AR) to assist in the targeted media delivery task and allow for preoperative image-derived models to be aligned with their intraoperative target site in the field of view of the surgeon. Augmented reality provides the ability to combine, register, and display interactive 3D virtual content in real-time [11]. By using a stereoscopic optical see-through head-mounted display (OST-HMD) to visualize computer-generated graphics, the visual disconnect between the information presented on a monitor and the surgical scene can be eliminated [12]. When effectively implemented, see-through HMD use, for surgical navigation, can provide enhanced visualization of intraprocedural target paths and structures [13].

### 1.2. Related Work

In the context of surgical navigation, see-through HMDs have been explored in neurosurgery [14], orthopedic surgery [15], minimally invasive surgeries (laparoscopic or endoscopic) [16], general surgery [13], and plastic surgery [17]; however, HMD led AR for targeted cardiac procedures, particularly therapeutic delivery, remains unexplored.

The perception location of augmented content and the impact of different visualization paradigms on user task-performance remains an active and ongoing area of research in the AR space. Using a marker-based alignment approach, the perceptual limitations of OST-HMDs, due to contribution of the vergence-accommodation conflict, have been investigated in guidance during manual tasks, with superimposed virtual content, using the HoloLens 1 [18] and Magic Leap One [19]. In a non-marker-based alignment strategy, other groups have focused on investigating how different visualization techniques or a multi-view AR experience can influence a users’ ability to perform perception-based manual alignment of virtual-to-real content for breast reconstruction surgery [20] or robot-assisted minimally invasive surgery [21].

Though there have been significant efforts going into the design of see-through HMD-based surgical navigation platforms, applications have remained primarily constrained to research lab environments and have experienced little clinical uptake—to date, there are no widely used commercial see-through HMD surgical navigation systems [22]. The historically poor clinical uptake of these technologies can be attributed to a lack of comfort and HMD performance [16], poor rendering resolution [23], limitations in perception due to the vergence-accommodation conflict [24], reliance upon a user to manually control the appearance and presentation of virtually augmented entities [25], and poor virtual model alignment with the scene due to failed per-user display calibration [18].

### 1.3. Objectives

Using an OST-HMD, our goal was to contribute to the initial design and assessment of an AR-based guidance system for targeted manual tasks and to demonstrate the capacity of our approach to display MRI-derived virtual models aligned with target tissue in a motion-arrested surgical model. The long-term goal of our work is to expand this into a standalone surgical navigation framework to guide the delivery of targeted media to the heart by the accurate alignment and visualization of MRI-derived virtual models with target tissue.

Through extensive user-driven experiments, we evaluated the comparative perceptual accuracy and usability of a monitor-led guidance approach against several paradigms, which used an OST-HMD for virtual guidance. Further, we presented and evaluated our efficient user-specific interaction-based display calibration technique for the OST-HMD to assess its contribution to performance in the user-study. In the context of our current work, perceptual accuracy refers to the quantitative assessment of a user’s interpretation and understanding of virtually augmented content. As an initial experiment to evaluate its clinical potential, we qualitatively assessed the capacity of our system, for displaying virtual MRI-derived scar models aligned with the heart, in an open-chest in-situ model of porcine MI. To facilitate collaboration and accelerate future developments in the AR community, we publicly released our software for OST-HMD calibration and alignment (https://github.com/doughtmw/display-calibration-hololens, accessed on 28 December 2021).

## 2. Materials and Methods

### 2.1. Display Device

We used the Microsoft HoloLens 2 (https://www.microsoft.com/en-us/hololens, accessed on 28 December 2021) OST-HMD for display of virtual models for guidance. The HoloLens 2 includes several cameras, an inertial measurement unit, accelerometer, gyroscope, and magnetometer, enabling head tracking in six-degrees of freedom (6DoF) without the requirement of an external tracking system. The HoloLens 2 is capable of the visualization of 3D virtual models, through stereoscopic vision, via two 2D laser-beam scanning displays, offering a field of view of 43 × 29 degrees (horizontal × vertical) to the wearer.

### 2.2. Virtual Model Alignment

#### 2.2.1. Sensor Calibration

We performed a camera calibration procedure of the red, green, and blue (RGB) HoloLens 2 photo-video camera sensor (896 × 504 px at 30 FPS). Intrinsic parameters (K C) such as focal length $\left(f_{x},\;f_{y}\right)$, principal points $\left(c_{x},c_{y}\right)$, and skew (s), as well as extrinsic rotation and translation parameters $\left({[}_{C}^{W}R\;|{\;}_{C}^{W}t\right])$, were estimated [26]. Together, these parameters described a perspective projection, which related points in world coordinates $\left(x_{W}\right)$ to their image projections $\left(u_{C}\right)$ (Figure 2a).

We adopted the same terminology as in [27]; bold lower-case letters denote vectors, such as 3D points xW∈ℝ3, or 2D image points uC∈ℝ2, whereas upper case letters denote matrices such as a rotation matrix R∈ℝ3×3. A 6DoF transformation from the world coordinate system (W) to the camera coordinate system (C) is described with ( CWR,  CWt), and a 3D point xW can be transformed from the world (W) to camera (C) coordinate system with xC= CWR xW+ CWt= CWT x¯W, where T∈ℝ4×4 is a 4×4 transformation matrix, and x¯W is the homogeneous (or projective) coordinate representation of vector xW.

#### 2.2.2. Per-User Hsead-Mounted Display Calibration

To compensate for discrepancies between the coordinate system of the now calibrated tracking sensor and the eye geometry of a user, a problem common to OST-HMDs [27], these spaces need to be aligned. Previous approaches to alignment have used manual [28], semi-automatic [29], and automatic [30] methods. Due to their relative ease of implementation and robust results [27], we incorporated a user-driven manual interaction-based calibration technique.

Manual approaches to display calibration have relied upon user interaction to collect point correspondences, typically requesting a user to align a tracked object in the world with a virtual target displayed on the screen of an OST-HMD. After a sufficient number of point correspondences have been collected, a transform is computed to relate these coordinate systems and complete the alignment process [28]. The HMD calibration procedure typically results in a 3×4 projection matrix to directly relate 3D head-relative camera points $\left(x_{C}\right)$ to 2D image points $\left(u_{E}\right)$; however, in some setups, it is not possible to modify these default internal projection matrices. To overcome this challenge, we followed an approach similar to Azimi et al. [31] and framed our calibration procedure as a black box 3D to 3D alignment between the head-relative camera points $\left(x_{C}\right)$ and the points in the projective virtual space $\left(x_{E}\right)$ (Figure 2b). 

During the alignment process, a user was tasked with aligning a tracked object $\left(x_{C}\right)$ with a virtual reticle in the projective virtual space $\left(x_{E}\right)$ (Figure 2c). A 4×4 transform for the left ( E,LCT) and right ( E,RCT) eyes was estimated using the collected 3D-3D point correspondences to optimally relate these coordinate systems. The default projection matrix configurations of the HMD for the left ( E,LCPdefault) and right ( E,RCPdefault) eyes were left unmodified; instead, we applied our transform to both internal projection matrices, resulting in the following relations for the left and right eyes of a user:(1) E,LCP= E,LCPdefault· E,LCT
(2) E,RCP= E,RCPdefault· E,RCT

The result was two 3×4 corrected projection matrices, adjusted to align virtual content with tracked objects for the left and right eyes of a user.

To estimate the camera to eye offset ( ECT) from the point correspondence data collected from user-interaction, we found that a rigid transformation provided an optimal balance of stability and accuracy. To arrive at a unique rigid transform solution, we required a minimum of three unique point correspondences. We approached the optimal rigid transform solution by minimizing the least-squares error as [32]:(3)error=∑i=0N‖Rai+t−bi‖2
where a and b are sets of 3D points with known correspondences, R is the 3×3 rotation matrix, and t is the translation vector. The optimal rigid transformation matrix was found by computing the centroids of both point datasets, shifting the data to its origin and using Singular Value Decomposition (SVD) to factorize the composed matrix [32]:(4)H=(a−centroida)(b−centroidb),H=USVT, X=UVT
where H∈ℝn×p is the accumulated matrix of points, U∈ℝn×n, S∈ℝn×p, and VT∈ℝp×p are the factorized components of H, and X is a 3×3 rotation matrix. Equation (4) is occasionally subject to reflection (mirroring); this case is handled through the incorporation of a determinant result d, enabling the estimation of the rotation as [33]:(5)d=determinant(X),R=U(10001000d)VT
where R is our final 3×3 rotation matrix. We computed the translation component as [32]:(6)t=centroidb−R·centroida
where t is the translation vector. We composed the camera to eye offset transform  ECT (for the left and right eyes) from the rotation matrix and translation vector components ([R | t]).

We found that a set of 10 point correspondences offered an optimal balance between the speed of calibration and calibration performance. For each point correspondence, the on-screen virtual reticle $\left(x_{E}\right)$ was moved to a predefined 3D position within the field-of-view of the user. The user was then tasked with aligning a tracked marker $\left(x_{C}\right)$ with the on-screen virtual reticle (Figure 2c) and pressing the space key on a connected keyboard (or performing the double-tap gesture) to save the point correspondence data from alignment.

#### 2.2.3. Marker Tracking

After calibrating the camera sensor and performing the per-user display calibration procedure, we aligned virtual content with the scene, using rigid-body registration, based on square-marker fiducials. Our software was implemented using Unity (https://unity.com/, accessed on 28 December 2021). We incorporated a Windows C++ Runtime Component in our Unity project that used the Eigen (https://eigen.tuxfamily.org/, accessed on 28 December 2021) library to implement the algorithm described above to compute the optimal rigid transform, from point correspondence data and the ArUco library [34], for the alignment of virtual models with square-marker fiducials.

### 2.3. User Study to Assess Perceptual Accuracy

To evaluate the perceptual accuracy of our approach, we designed a set of point-and-trace tasks for users to perform, and we compared their performance across four guidance paradigms.

#### 2.3.1. Study Details

We began by guiding users through the flow of the user-study tasks and describing each individual task paradigm. Next, we introduced users to the HoloLens 2 and assisted them with fitting and performing the pupillary calibration procedure included with the device. The pupillary calibration procedure was used to estimate the interpupillary distance (IPD) of the wearer and was performed once, for each user in the study, prior to beginning any of the assessment tasks.

We then allowed users time to familiarize themselves with the HoloLens 2 system and the associated gestures for interaction. Finally, users were tasked with performing the trace test for comparing each of the guidance paradigms and providing feedback of their experience. The comparison study required one hour per user.

We asked users to return one week following the initial comparison study for a repeatability assessment. Prior to beginning the assessment tasks, participants were once again asked to perform the HoloLens 2 pupillary calibration to estimate their IPD, then repeat a specific test paradigm twice, and provide feedback for each trial. The repeatability assessment trial required an additional 30 min per user. Instead of removing the headset after each task, users were instructed to flip the HoloLens 2 visor up to ensure the headset did not shift substantially during testing.

#### 2.3.2. Display Paradigms

For the comparison study, four visualization paradigms were tested across four separate tasks. The ground truth and point-and-trace model was: (1) displayed with its real-world scaling on a monitor placed near the user ($T_{M}$) (Figure 3a); (2) displayed adjacent to the trace template via the HMD ($T_{A}$) (Figure 3b); (3) registered directly to the trace template using ArUco markers and displayed via the HMD ($T_{D}$) (Figure 3c); (4) registered directly to the trace template, using ArUco markers and displayed via the HMD, after performing per-user display calibration ($T_{C}$) (Figure 3c).

In the repeatability assessment, users performed the point-and-trace task after per-user display calibration, while guided by the HMD, a total of two times ($T_{R}$; the same setup as in $T_{C}$ described above) (Figure 3c). We hypothesized that, following a repeat trial, the users would be more comfortable with the display calibration procedure, leading to better calibration results, less time needed for calibration, and reduced task-load.

#### 2.3.3. Point-and-Trace Tasks

Using representative simulated 2D scar morphology of similar irregular shape and scale, we performed a series of user-driven 2D point-and-trace tests. Users were tasked with delineating scar boundaries and identifying mock injection point locations on a sheet of paper, while guided by each of the proposed visualization paradigms, to evaluate the accuracy and time required to complete the task. Figure 4a includes a sample scar ground truth template. Figure 4b shows a blank sample trace template, and Figure 4c demonstrates a sample scanned user-trace result. A total of 19 simulated injection locations, spaced roughly 1 cm apart, were included to mimic the constraints of a typical media-delivery procedure in a porcine model [8]. To allow for relation of user-drawn points with their respective ground truths, we noted the order in which the user approached the simulated injection task during user-testing.

For the experiment, we created a set of 5 unique ground truth phantoms, which were variations of the ground truth template presented in Figure 4a, with different contour shapes and point locations but the same total number of points. During testing, the task order for user-testing and the ground truth trace template presented to the user were randomly selected to ensure that there was minimal contribution of user-learning to performance in each visualization paradigm. The trace template provided to users was the same for each test and included minimal information to assist in localization, such as simulated vessel landmarks and an outer contour to indicate the extent of the heart (Figure 4b).

#### 2.3.4. Metrics for Evaluation

To facilitate the desired 1 cm spacing of injection sites for targeted media delivery to the heart, we aimed to achieve a target registration error (TRE) of less than 5 mm. The TRE goal of 5 mm was selected to minimize the potential overlap of delivered media, as a result of mistargeted injections, and to account for typical diffusion patterns in tissue post-injection [8]. Additionally, to minimize the interruption to a typical workflow, we set out to complete the calibration procedure in less than 5 min. We identified that a 5-min time threshold for system calibration and setup would fit comfortably within the existing workflow of media delivery procedures and could be performed while pre-procedural patient preparation and workspace setup was being performed.

Target registration error informs the overall functional accuracy of the guidance system, not corrected by any rigid translation, and it is dependent on the uncertainties associated with each of the individual components in the guidance application [35]. The TRE can be formulated as: (7)$$\mathrm{TRE}=d\left(p,q\right)=\sqrt{\sum_{i=1}^{N}\left(p_{i}-q_{i}\right)}$$
where d(p,q) is the Euclidean distance between a spatial position of interest in surgical space (p), and the corresponding spatial position in OST-HMD view space after registration and alignment has been performed (q) [36]. For the point-and-trace test, we evaluated the TRE between the centroids of ground truth and user-drawn point locations as in Figure 4d.

The average symmetric surface distance (ASSD) was used to measure the distance discrepancy between the ground truth scar boundary trace and the user-drawn scar boundary, as in Figure 4e. We defined the ASSD metric as:(8)$$\mathrm{ASSD}=\frac{\sum_{x\in \partial G}d\left(x,\partial M\right)+\sum_{y\in \partial M}d\left(y,\partial G\right)}{\left|\partial G\right|+\left|\partial M\right|}$$
where d(x,∂M) is the distance between a border pixel of the ground truth contour (x) and the nearest user-drawn contour (∂M) pixel (reverse for d(y,∂G)). The summation is performed across all pixels of the ground truth (∂G) and the user-drawn contours (∂M) to compute an average distance metric in millimeters. During each paradigm, we measured the task completion time and the time required for completing the per-user display calibration procedure.

Subjective evaluation was performed after each paradigm of the point-and-trace user-study through the self-reported NASA task load index (NASA-TLX) [37]; users were instructed to include their assessment of the additional display calibration procedure when performing the NASA-TLX. We used a scanner to digitize the information from each user trial at a resolution of 300 dots per inch (DPI). MATLAB R2020a (https://www.mathworks.com/products/matlab.html, accessed on 28 December 2021) was used to automatically align the ground truth targets with scanned user-traces, based on an intensity-based registration algorithm, resulting in a 2D rigid transformation with 3 degrees-of-freedom. After each automated registration, correct alignment of the scanned ground truth trace was confirmed with the scanned user trace by visual assessment of the overlap of the four square-marker fiducials surrounding the template and user trace (visible in Figure 3). After confirming alignment, we used MATLAB to calculate the ASSD and TRE metrics from the aligned image pairs.

#### 2.3.5. Statistical Analyses

We used the Shapiro–Wilk parametric hypothesis test to assess the normality of the quantitative metrics measured; the test indicated that the null hypothesis of composite normality was a reasonable assumption [38]. One-way analysis of variance (ANOVA) was performed to determine if data from the tested conditions had a common mean: a threshold of p<0.05 was set to indicate a statistically significant difference. We used the Bonferroni correction to adjust for multiple comparisons across the test paradigms.

### 2.4. Feasability Assessment in a Porcine Model

All animal procedures were approved by, and were in accordance with, the Animal Care Committee at Sunnybrook Research Institute. Myocardial infarction was induced in two adult pigs through a 90-min complete balloon occlusion of the left anterior descending (LAD) artery, followed by reperfusion; this is known to create large transmural anteroseptal wall scars [10].

#### 2.4.1. Virtual Scar Model

To create a high-resolution model of the heart and scar tissue, we included cine and LGE imaging in our MRI scan protocol. Cine imaging is a bright-blood technique based on a steady-state free precession gradient echo sequence that allows for rapid imaging across phases of the cardiac cycle, assessment of ventricular volumes, and ejection fraction [39]. We use gadolinium diethylenetriamine penta-acetic acid (Gd-DTPA), a paramagnetic contrast agent, to assess myocardial perfusion and structural changes in the extracellular space [39]. Late gadolinium-enhanced (LGE) imaging is a “T1-weighted” technique that quantifies scarring post-MI.

Prior to the imaging protocol, MRI visible fiducial markers were applied to the animal in a reproducible configuration. Using 3D Slicer [40], segmentations of left ventricle myocardium and scar tissue were created from the MRI data, resulting in surface models of each structure. Model decimation and Laplacian smoothing was performed on input segmentations in Blender (https://www.blender.org/, accessed on 28 December 2021) and provided smoothed voxel-wise 3D models of cardiac anatomy. To avoid occlusion of underlying anatomy, we used transparent surface rendering for virtual model visualization. The 3D locations of MRI-visible fiducials were extracted from image-space for the configuration of the ArUco tracking object in Unity.

#### 2.4.2. In-Situ Visualization

At 12 weeks post-MI, two porcine models underwent a left lateral thoracotomy procedure to expose the motion arrested in-situ heart. To ensure that there were minimal anatomical changes due to animal growth, the MRI scan protocol was performed the same day as the sacrifice and thoracotomy procedure. Fiducial landmark locations from imaging were repeated by placing the square-marker fiducials in an identical configuration intraoperatively, permitting rigid registration of image-derived virtual models from imaging to intraoperative space, based on the precise positioning of these square-marker fiducials. After performing our per-user HMD calibration procedure, we qualitatively assessed the alignment of the MRI-derived 3D scar models with the surface of the heart.

## 3. Results

### 3.1. User Study Results

Across the test paradigms, we evaluated and compared the TRE (Euclidean error), RMS TRE $\left(\mathrm{RMSE}=\sqrt{\mu^{2}+\sigma^{2}}\right)$, ASSD, time-to-task completion, and calibration time. 

#### 3.1.1. User Details

We recruited a total of 10 users, from the University of Toronto and Sunnybrook Research Institute, with little to no experience with AR for the study; detailed statistics on the participants are included in Table 1.

#### 3.1.2. Point Accuracy

The most critical metric in the evaluation of our guidance framework is error. We included the measures of TRE for the comparison and repeatability assessments in Figure 5a. In the monitor-guided paradigm ($T_{M}$), the TRE was 3.4±2.2 mm (mean ± standard deviation) (${\mathrm{RMSE}}_{M}=4.03$ mm). The HMD guided paradigm, where virtual content was shown adjacent to the trace target ($T_{A}$), gave a TRE of 2.6±1.4 mm (${\mathrm{RMSE}}_{A}=3.00$ mm). The HMD guided paradigm, prior to per-user display calibration with direct alignment of virtual content ($T_{D}$), resulted in a TRE of 1.1±0.6 mm (${\mathrm{RMSE}}_{D}=1.26$ mm), and after display calibration ($T_{C}$) a TRE of 1.02±0.6 mm (${\mathrm{RMSE}}_{C}=1.17$ mm). In the repeatability study ($T_{R}$), we measured a TRE of 0.98±0.5 mm (${\mathrm{RMSE}}_{R}=1.10$ mm). 

We measured a significant improvement in the mean TRE of the HoloLens 2 guided paradigms with direct guidance ($T_{D}$, $T_{C}$, and $T_{R}$) when compared to the monitor-guided ($T_{M}$) and HoloLens 2 adjacent guided ($T_{A}$) paradigms (p<0.05). With our repeat testing session, we observed similar accuracy as we did in the initial comparison study session, indicating that the novice users were able to reach maximal performance in the first evaluation. 

#### 3.1.3. Contour Tracing

For our application of targeted media delivery to the heart, the precise localization of contour boundaries of scar myocardium is essential to minimize the risk of mistargeting and delivering media to healthy regions of myocardium. In Figure 5b, we report the ASSD measures for contour tracing in the comparison and repeatability assessments. The monitor-guided case ($T_{M}$) resulted in an ASSD of 1.75±0.64 mm, with an ASSD of 1.15±0.40 mm for the HMD led case with adjacent virtual content ($T_{A}$). In the HMD direct visualization paradigms, an ASSD of 0.51±0.12 mm was measured prior to display calibration ($T_{D}$) and 0.50±0.11 mm after per-user display calibration ($T_{C}$). In the repeat assessment ($T_{R}$), we recorded an ASSD score of 0.51±0.11 mm.

Like the point accuracy assessment, we measured a significant improvement in the ASSD of the HMD direct, direct calibrated, and direct calibrated repeat paradigms ($T_{D}$, $T_{C}$, and ) when compared to the monitor-guided ($T_{M}$) and HMD adjacent guided ($T_{A}$) paradigms (p<0.05). 

#### 3.1.4. Task Completion Time

In Figure 5c, the results of task completion time are compared across the test scenarios. The time required for point collection was extracted as the calibration time. For the monitor-guided paradigm ($T_{M}$), we measured 129±72 s for completion of the point-and-trace task, with 130±63 s for the HMD guided case with adjacent virtual content ($T_{A}$). In the HMD direct visualization paradigms, users required 78±34 s for task completion prior to display calibration ($T_{D}$) and 71±23 s, after performing per-user display calibration ($T_{C}$). In the repeat assessment, we recorded a task completion time of 73±29 s ($T_{R}$).

We observed a non-significant reduction in the time required to complete the point-and-trace task when the virtual ground truth target was aligned directly with the target ($T_{D}$, $T_{C}$, and $T_{R}$).

#### 3.1.5. Calibration Time

The time required for user-driven display calibration, during the comparison and repeatability assessments, is included in Figure 5d. We observed a non-significant reduction in calibration time between the initial trial 122±65 s ($T_{C}$) and the repeat trials 85±38 s ($T_{R}$). The reduced time required for calibration in the repeat trial could indicate that users felt more familiar with the calibration procedure in their return visit.

#### 3.1.6. NASA-TLX Results

The results of the raw (unweighted) NASA-TLX are included in Figure 6 and serve as a subjective assessment of the perceived task load. We measured a significant reduction in perceived mental demand and required effort in the HoloLens 2 direct visualization paradigm ($T_{D}$). When incorporating the additional per-user display calibration procedure ($T_{C}$, $T_{R}$), the user-perceived performance displayed a significant improvement; however, this came at the expense of increased mental and physical demand, as well as perceived user-effort. 

At the end of each session, we asked participants if they experienced any nausea/dizziness, eye strain, or neck discomfort. One user reported eye strain during the calibration procedure due to discomfort with closing their left and right eyes while viewing the calibration reticle. No other reports of discomfort were reported by the participants.

### 3.2. Porcine Study Results

We assessed the feasibility of our proposed HMD based guidance framework with per-user calibration for visualization in two in-situ porcine models. Figure 7a contains a slice of the LGE MRI data and segmentations, with Figure 7b showing the corresponding surface-rendered virtual models of tissue structures and volume rendered locations of MRI-visible fiducials. Figure 7c–e include images from the HoloLens 2 mixed reality capture, showing the exposed surface of the porcine heart with its aligned image-derived virtual model. After achieving initial alignment using the marker-based approach, simultaneous localization and mapping (SLAM) was used to maintain registration of the virtual model for the duration of the experiment (due to the shift in tissue and marker locations during the thoracotomy). Overall, there was good agreement between the 3D virtual scar model and the actual scar (and minimal registration drift throughout), and the model appeared well in the surgical lighting conditions. Averaged across the two porcine studies, we found that the fiducial marker placement procedure required 163.2±22.6 s, and the per-user display calibration procedure needed 118.9±26.3 s, placing us within our goal of a sub-5-min setup and calibration time at 4.7 min.

## 4. Discussion

To our knowledge, we are the first to focus on investigating the perceptual accuracy of augmented virtual content using the HoloLens 2 headset and to compare its performance to a conventional monitor-guided approach. In similar user-driven assessments of perceptual accuracy, Qian et al. reported a Euclidean error of 2.17 mm RMS using the Magic Leap One OST-HMD [19], and Ballestin et al. reported a 2D reprojection error of 9.6 mm RMS using the Meta2 OST-HMD [41]. Both approaches used a single-point active alignment calibration procedure (SPAAM) to estimate a 3D to 2D projective transform. In earlier work, Qian et al. used a 3D-to-3D display calibration technique and arrived at a reprojection error of 2.24 mm RMS on the HL1 and 2.76 mm RMS using the Epson Moverio BT-300 [31].

In our study, we recorded significantly increased error with a conventional monitor-guided approach when compared to HMD-led techniques that directly superimposed virtual content on the target site. This disparity in performance was especially noticeable across target points that were far away from landmark features in the ground truth trace template. For similar guidance tasks as our 2D point-and-trace experiment, we have concluded that a monitor-based guidance approach is sub-optimal and, in our case, lead to an RMSE of 4 mm.

The proposed HMD-led adjacent visualization resulted in significantly increased error (3 mm RMSE) when compared to HMD-led guidance with direct superimposition. Though the HMD-led adjacent paradigm brought the augmented virtual guidance into the field-of-view of the wearer during task performance, it did not significantly outperform the monitor-guided task. From this result, we gather that, to achieve the maximal accuracy in a similar task, the ground truth information needs to be directly superimposed with the target site to remove the requirement of users to rely on a mental mapping and features in the image for orientation and guidance.

In our experiments with direct alignment of the virtual ground truth with the target site, we recorded error of 1.26 mm RMSE without additional display calibration and error of 1.1 mm RMSE after our manual per-user calibration approach. We believe that this minimal improvement to accuracy is partially due to the contribution of active corrections made to virtual content augmentation, via active eye-tracking and improved IPD estimation, on the HoloLens 2. To test this hypothesis, we assessed the performance of our calibration approach on the HoloLens 1 (HL1), an OST-HMD which does not benefit from eye-tracking. With a smaller subgroup of participants (N=3) we re-evaluated the point-and-trace task using the HL1. In the direct visualization paradigm ($T_{D,\;HL1}$), we measured a TRE of 3.5±1.1 mm (${\mathrm{RMSE}}_{D,\;HL1}=3.64$ mm) and, including additional manual per-user calibration ($T_{C,\;HL1}$), a TRE of 1.8±0.6 mm (${\mathrm{RMSE}}_{C,\;HL1}=1.89$ mm). From this experiment, we gather that the incremental improvement to perceptual accuracy, with the inclusion of an additional manual display calibration procedure, was less with the HoloLens 2 (13% improvement) than the HL1 (48% improvement). We can also extract that the error reduction during the task, due to the active eye-tracking and display technology of the HoloLens 2 versus the HL1, was on the order of 0.65 mm.

As OST-HMD technology continues to mature, the development direction seems to point to the use of a lightweight headset and comfortable form factor, with many of the cumbersome on-board computing resources removed in favor of a shift to cloud-based computing. We anticipate that these lightweight headsets, which may not include eye-tracking, will benefit from the proposed efficient per-user display calibration procedure. However, we suggest that researchers must evaluate the requirements of their proposed application space to assess whether the improvements to perceptual accuracy are worth the trade-off of potential cognitive loading and the additional time required for manual display calibration.

To minimize the time interruption to a typical media delivery procedure, we aimed for the OST-HMD display calibration procedure to require less than 5 min; our data demonstrates that this is feasible, as we required 1.42 min for calibration on average. In recent work by Long et al., the authors investigated the perceptual accuracy of a Magic Leap One headset with affixed dental loupes for enhanced magnification. In the AR-Loupe condition, users achieved 0.94 mm RMS error (compared the 1.10 mm RMS in our study) but required 4.47 min, on average, for the point alignment task [19]—more than triple the 1.42 min required in our study.

The vergence accommodation conflict is commonly discussed when evaluating the limitations of current available commercial OST-HMDs and stems from the difference in distance between the depth of targets in the scene (vergence) and fixed focal distance of the OST-HMD (accommodation). This conflict can lead to discomfort to users due to out of focus images and reduced perceptual accuracy of virtual content—especially in the depth plane [24]. At a focal distance of 0.5 m, depth of field predictions range from 0.40 to 0.67 m, which fit well within the peri-personal space requirements [18]. However, the HoloLens 2 has a fixed focal distance of 1 m to 1.5 m and is designed for an optimal user experience in the range of 1 m to 2 m viewing distance, resulting in mismatched vergence accommodation for virtual content in the peri-personal space and the potential for errors in perception.

To categorize errors due to marker tracking, we defined misalignments of ground truth landmarks, with measured ArUco fiducial points, as the RMS fiducial registration error (FRE):(9)$$\mathrm{FRE}=\sqrt{\left(\frac{1}{N}\right)\overset{N}{\underset{i=1}{\sum }}{\left(m_{i}-n_{i}\right)}^{2}}$$
where m are the centroids of ground truth landmarks and n are the centroids of physically measured fiducial points (N total points) after performing registration between these spaces. Using the ArUco library, experimental measures of FRE range from 0.27–0.91 mm for 40 cm and 100 cm viewing distances, respectively, with rotational errors between 1.0–3.8 degrees [42]. By combining the contributions from ArUco marker tracking, we expect a similar FRE of 0.27–0.91 mm for our proposed viewing distance. As the FRE contributes to the total TRE, we can expect a TRE in the order of 0.3–0.9 mm to be the theoretical best accuracy of our system—assuming that there are no other contributing sources of error in the registration due to manual calibration and the intrinsic display properties of the OST-HMD. Comparing the best-case assessment of TRE, with 0.3–0.9 mm of error, contributed, based on marker tracking alone, to the TRE of 1.1 mm RMS measured in our repeatability study ($T_{R}$), we can infer that the error, due to other sources such as display calibration and vergence-accommodation, is in the order of 0.2–0.8 mm. 

With registration of imaging data from preoperative imaging to intraoperative surgical space, we expect additional error on the order of 0.2 mm [43] due to the inherent resolution limitations of MRI. Therefore, the theoretical best-case TRE of our combined guidance system is 1.35 mm RMS. Importantly, the total combined TRE of our approach is small, relative to the contribution of expected cardiac and respiratory motion error at the target site. At the LAD, the typical displacement has been measured as 12.83 mm RMS without, or 1.51 mm RMS with, mechanical stabilization in healthy porcine models [44].

The registration approach used in the porcine study presents significant potential for user-introduced error through the requirement for manual re-placement of square image-based markers onto the animal, intraoperatively, in an identical configuration as the preoperative MRI-visible fiducials. Given that the user-study was focused on a marker-based registration led assessment of perceptual accuracy in a 2D task, we felt that introducing a new registration paradigm, which is less sensitive to user-introduced error for the in-situ porcine assessment, could confuse the reader and take away from the validity of this assessment. In our approach to registration, we pre-constructed the marker-based tracking configuration with knowledge of the expected relative marker poses, as measured from preoperative imaging space and the MRI-visible fiducials. Through pre-construction of this tracking object, our approach was more robust to occlusion and only required concurrent detection of two out of four markers to maintain alignment [34].

As discussed in the porcine study results section, we mention that the marker-based tracking approach was only used to achieve an initial registration at the beginning of the procedure, after which we relied on the SLAM capabilities included with the HoloLens 2 for ongoing alignment. We found that, due to the significant shift in tissue and marker locations during surgical entry and access to the surface of the heart, SLAM tracking was more reliable through the duration of the procedure—an indicator, to us, of a potential area for improvement with future work. An alternative and intermediate step, prior to moving fully towards a surface-based registration approach such as point-cloud matching, is to investigate the use of a magnetic ink tattoo, which is visible in MRI, as a fiducial marker for RGB based tracking [45], to enable more precise preoperative imaging to intraoperative surgical alignment without requiring the manual re-placement of fiducials.

We acknowledge that there is a substantial abstraction in task difficulty when moving from the 2D point-and-trace user study to the 3D feasibility assessment in an in-situ porcine model; however, for our specific proposed use case of targeted media delivery to the epicardial surface of the infarcted heart, we feel that the 2D approximation of target tissue is acceptable for the following reasons.

Though the heart itself is inherently of complex 3D morphology, for a targeted delivery task, we are primarily focused on the correct 2D placement and 2D spacing of media injections across a section of the epicardial surface of the heart [8]—specifically, the scar myocardium identified through LGE MRI. After balloon occlusion of the LAD, the resulting anteroseptal wall scar tissue typically appears as a relatively flat section across the surface of the heart with no extreme curvature (Figure 1a). The morphology of the scar tissue is also shown in the accompanying LGE MRI data (Figure 1b).

Importantly, the media delivery plan (Figure 1c), which serves as the current reference for intraoperative media placement guidance, is created as a 2D abstraction of the 3D infarct tissue on the surface of the heart. Currently the challenge for the surgeon is to localize and project a mental mapping of the 2D media delivery plan onto the surface of the heart, not to understand or interpret the 3D morphology of the characterized infarct.

In the chronic stage of MI (>4 weeks), the transmural (complete wall thickness) myocardial scar tissue has typically experienced significant regional wall thinning, relative to healthy myocardium, with wall thickness measurements of 5–6 mm [10], meaning that, during media delivery, there is little consideration as to the depth of injection into the scar tissue surface, apart from avoiding accidental perforation into the heart. As such, the focus of our efforts was on guiding the correct 2D localized placement of media injections into scar tissue rather than considering the 3D depth of media delivery.

We believe in the validity of the 2D approximation of 3D anatomical structures that was incorporated for our task-specific use case; however, this approximation does not necessarily apply for other navigation tasks, which require more complex 3D guidance. In that case, we would need to validate our system through a rigorous 3D phantom study to assess its capacity in 3D guidance and localization.

In our simplified scenario, we sought to demonstrate the capability of our system to display MRI-derived virtual models, aligned with target tissue, and to evaluate its capacity to be set up within the time constraints of a typical workflow. The in-depth quantitative assessment of the accuracy of our system in a complex clinical setting is beyond the scope of this work. Additionally, there are several other fundamental challenges to address, such as cardiac and respiratory motion, before attempting to evaluate a standalone OST-HMD based surgical navigation framework.

## 5. Conclusions

In this work, we measured the perceptual accuracy of several different guidance paradigms through an extensive user-study and demonstrated the potential for their use in displaying MRI-derived virtual models, aligned with myocardial scar tissue, in an in-situ porcine model. Our results indicated that the use of the HoloLens 2 OST-HMD and direct alignment of the virtual ground truth with the target site (with an additional display calibration procedure that required 1.42 min on average) enabled users to achieve a TRE of 1.1 mm RMS, as compared to a TRE of 4.03 mm RMS in the monitor-led case, in a simplified 2D scenario, as evaluated by the point-and-trace task. We observed that guidance paradigms, where the virtual ground truth was directly aligned with the target site, provided significant improvements to accuracy, contour tracing performance, and user-perceived performance, as measured by the NASA-TLX (p<0.05). In the porcine study, we demonstrated the potential capabilities of this system to display MRI-derived virtual content aligned with target tissue, moving us closer to the long-term goal of assisting in the delivery of media to target regions of scar tissue on the surface of an infarcted porcine heart.

We envision that our OST-HMD based navigation framework will offer a powerful platform for researchers, providing the means for improved accuracy during 2D targeted tasks and, with future developments, potentially contribute to optimal therapeutics and improved patient outcome. Furthermore, due to its utility and ease of incorporation, we expect that our navigation approach could have immediate implications in many other interventions, which could benefit from the accurate alignment and visualization of high-resolution imaging derived anatomical models.

## Figures and Tables

**Figure 1 jimaging-08-00033-f001:**
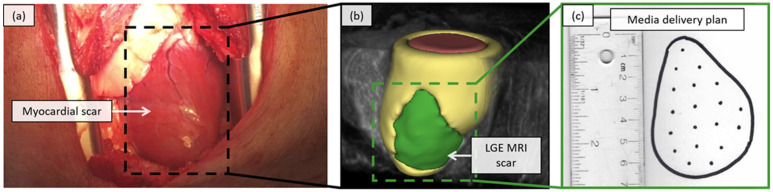
(**a**) Open chest surgical scene from a left lateral thoracotomy procedure, for delivery of a candidate therapy, in a porcine model of myocardial infarction, with (**b**) data from late gadolinium-enhanced magnetic resonance imaging (LGE MRI) to characterize myocardial scar tissue (green region), and (**c**) an example markup of a targeted media injection pattern, for the delivery of a candidate therapy, across the identified scar (target injection sites spaced 1 cm apart).

**Figure 2 jimaging-08-00033-f002:**
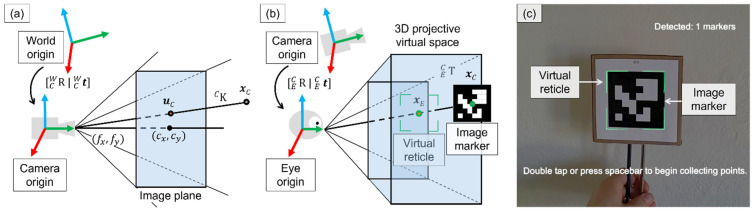
(**a**) Pinhole camera model (perspective projection) to relate 3D world points $x_{c}$ to their 2D image projections $u_{C}$ by the intrinsic matrix K C. Focal lengths length $f_{x}$ and $f_{y}$ are the distance between the pinhole and the image plane. Principal points are indicated as $c_{x}$ and $c_{y}\;$ [27]. (**b**) Physical viewing frustum (user perspective). A 3D rigid transform TEC is computed to relate 3D head-relative points of tracked content in the camera coordinate system $x_{c}$ to 3D viewpoints within the display of the HMD $x_{E}$ [27]. (**c**) Image captured through the head-mounted display during the display calibration procedure. The user is tasked with alignment of the tracked image marker (with centroid $x_{C}$ ) with the virtual on-screen reticle (with centroid $x_{E}$ ).

**Figure 3 jimaging-08-00033-f003:**
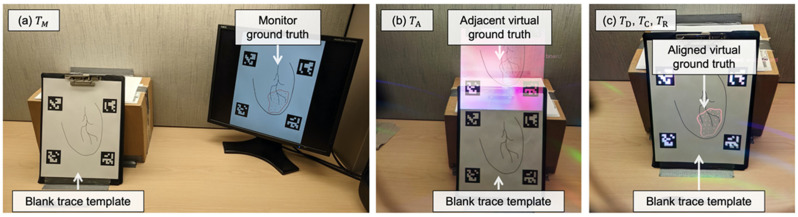
The display paradigms used in this study. (**a**) The monitor-guided condition ($T_{M}$) where the ground truth contour and point locations were displayed (with correct scaling) on a monitor adjacent to the user. (**b**) The HoloLens 2 guided condition ($T_{A}$ ), recorded through the HoloLens 2 display, with virtual content shown adjacent to the trace template and with the correct scale. (**c**) The HoloLens 2 guided conditions, recorded through the HoloLens 2 display, with ($T_{C}$, $T_{R}$ ) or without ($T_{D}$ ) performing per-user display calibration, where the ground truth template was directly aligned with the trace template and displayed to the user. In all paradigms, the user was provided with the same trace template that incorporates a minimal amount of feature information to assist in the point-and-trace task.

**Figure 4 jimaging-08-00033-f004:**
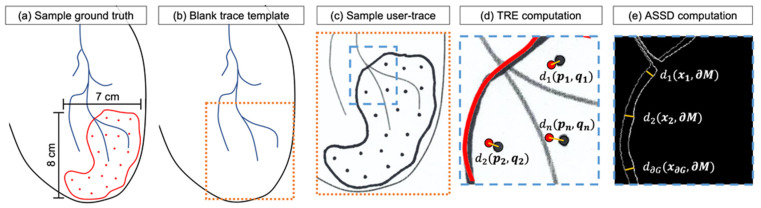
(**a**) A sample ground truth target showing the contour and target points in red. (**b**) A sample trace template and (**c**) scanned user trace result. (**d**) A sample target registration error (TRE) calculation, with circles fit to user-trace and ground truth points; (**e**) an average symmetric surface distance (ASSD) computation.

**Figure 5 jimaging-08-00033-f005:**
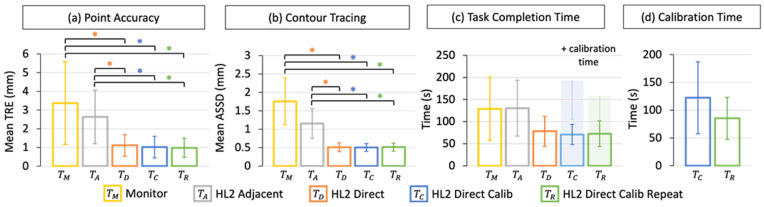
Quantitative results of the user-study. (**a**) Point accuracy was assessed as the mean target registration error (TRE), in millimeters, between user-points and ground truth points. (**b**) Contour tracing was measured as the average symmetric surface distance (ASSD), in millimeters, between the user and ground truth trace. (**c**) Task completion time and (**d**) per-user manual display calibration time were measured in seconds. HoloLens 2 (HL2) tasks are labelled. Significant results (p<0.05) are indicated by an Asterix (*).

**Figure 6 jimaging-08-00033-f006:**
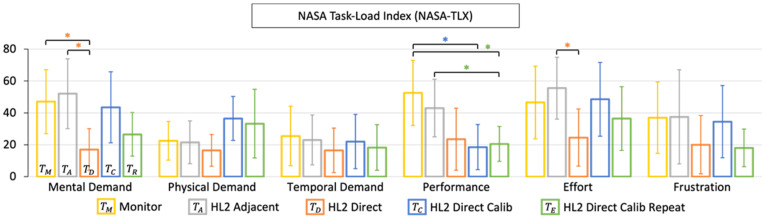
Qualitative results of the user-study. We report the results of the self-reported NASA task load index (NASA-TLX) scoring, which was performed after each paradigm of the point-and-trace user-study. HoloLens 2 (HL2) tasks are labelled. Significant results (p<0.05) are indicated by an Asterix (*).

**Figure 7 jimaging-08-00033-f007:**
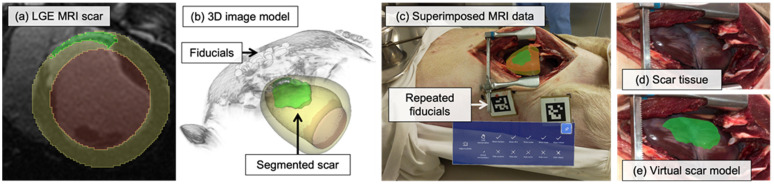
Results from the porcine feasibility study. (**a**) Segmented left ventricle myocardium and blood pool structures from cine MRI (yellow, red) and myocardial scar tissue from LGE MRI (green) at the 30-min timepoint after Gd injection. (**b**) Structures segmented from MRI as a smoothed 3D model with transparency applied. The MRI visible fiducials (Vitamin E caplets) are visible on the skin of the animal. (**c**–**e**) A single 2D view of the stereo 3D view (which a user wearing the HMD would see) was captured using the HoloLens 2 mixed reality capture capability, which is known to introduce an offset in the position of virtual models. The virtual model is shown aligned with the heart, and a user-interface is included to allow for control of virtual model mesh visualization. All images were recorded after animal sacrifice, resulting in higher scar to myocardium contrast from blood washout.

**Table 1 jimaging-08-00033-t001:** Participant details in the point-and-trace user-study.

Participants	Sex	Age (Years)	Glasses	Experience with AR
10	F: 4 , M: 6	33.6±11.4	5	None: 8, Minimal: 2

## Data Availability

The described software is available here: https://github.com/doughtmw/display-calibration-hololens (accessed on 28 December 2021). Additional data is available on request from the corresponding author.

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
