# Peer review of "Head-Mounted Display-Based Augmented Reality for Image-Guided Media Delivery to the Heart: A Preliminary Investigation of Perceptual Accuracy"

_2313-433X, 2022, doi:10.3390/jimaging8020033_

Round 1

Reviewer 1 Report

This study presents a framework for feasibility assessment of HMD-based augmented reality for use in heart surgery. The quality of writing and scientific presentation of the content is of very high quality. The presented techniques for evaluating the proposed visualization paradigms are sound and well explained. Although there is considerable value in comparing the different visualization schemes, I believe there are several factors that need to be carefully addressed/acknowledged in order for the presented methods to be of clinical relevance.

  1. The results and the methods presented in this paper are closely tied to a particular intervention, this should be reflected in the article's title.
  2. The is an underlying assumption throughout this article that a simplified 2D physical space and the corresponding 2D AR visualization can be a representative of a downstream 3D environment. This is not a valid assumption for many anatomies and surgical interventions where a complex 3D morphology is to be operated on. Although this is acknowledged by the authors in the discussion section as a limitation, the viability of the results presented in the herein article is heavily influenced by this assumption.
  3. Another simplification made is the availability of registration between the MRI space and the anatomy. Due to the above-mentioned assumption of a 2D target space, the registration is mainly derived based on tracked markers that are visible in both spaced. This of course is not a valid assumption for a realistic environment where no correspondence between the imaging space and the anatomy is available. This is particularly important for the visualization paradigm c (concluded to be the best visualization method), where almost a best-case scenario for registration is considered, therefore ignoring the potential confounding factors that may arise when a clinically plausible registration method (i.e., based on point cloud matching) is considered.
  4. Although the porcine study is noteworthy, these experiments appear to have resulted only in qualitative evaluation (e.g., line 423: there was good agreement …). Furthermore, registration is achieved based on MRI observed fiducial markers, which is not a clinically viable option. Therefore, it is hard to justify this experiment within the scope of the herein article.
  5. Section 2.2.1: please provide a list of the known and the unknown parameters. Do the extrinsic parameters of the HMD change for each calibration round? It is difficult to ascertain the estimated calibration parameters after through this calibration routine.
  6. Line 164: there appears to be a typo in E,LD should be E,L.
  7. There are multiple references to point tracing and guidance paradigms in section 2.3.1 without prior explanation (they appear later in 2.3.2 ad 2.3.3).
  8. Given that the target media delivery points are spaced 1cm apart, would you please elaborate further on the TRE goal of 5 mm, which is 50% of the point spacing?
  9. Line 293: how is the automatic alignment performed? How many parameters are estimated? Is there a rotation applied? Does this resolve some of the misalignments between the user drawn traces and the ground truth?

Author Response

Please see the attached word document.

Reviewer 2 Report

This manuscript technically and ergonomically compared the efficiencies (accuracy, speed and subjective workload, etc.) of OST-HMD-based AR in point-and-trace tasks of the targeted cardiac procedure with four different visualization-guidance paradigms respectively. Authors presented two kinds of experiments (the simulated surgical targeting and the motion-arrested open-chest porcine model) and concluded an improved perceptual accuracy and task-completion times in a simulated targeting task utilizing an OST-HMD. The following suggestions should be considered:

  1. The perceptual accuracy of OST-HMD-based AR directly related with the specific therapy indication from ergonomical consideration. As the myocardial infarction is selected by author to investigate the perceptual accuracy, it is better to add the therapy indication into the manuscript’s title. After all, author also claimed that “HMD led AR for targeted cardiac procedures, particularly therapeutic delivery, remains unexplored” in lines 74 and 75.
  2. The definition and related parameters of “perceptual accuracy” should be explicitly and clearly described when this phrase appears firstly in the line 104.
  3. For the equation in line 134, are two “xw” same? It seems that the first one is in the form of homogeneous coordinates, and the second one is in the form of un-homogeneous coordinates. Pls check and make an explanation if necessary.
  4. In Fig. 2 (a) and (b), it is hard to distinguish whether the plane or the origin-point is in the front. It is better to use solid or dotted line (segment) to represent the front/rear relationships of different plane/origin-points.
  5. In the symbol “left (CE,LDPdefault) in line 164, the character ‘D” maybe should be deleted. Pls check it.
  6. Author used “point-and-trace task”, “trace-and-point task”, “point-and-trace test”, “point-and-trace test”, “trace-and-point model”, “trace-and-point experiment”, etc. What is the correct order of “point” and “trace” in theses phrases.
  7. For the scanning of “total of 19 simulated injection locations” in line 245, does the location scanning order have influence on the efficiency of point-and-trace test?
  8. It is not clearly explained in line 264 that why “set out to complete the calibration procedure in less than 5 minutes”? Is it a standard to set 5 minutes in maximum?
  9. In line 292, which resolution is used in the scanner here?
  10. In line 326, “2.4.1” should be renamed “2.4.2”. Be careful pls.
  11. Table 1 used ‘N’ to denote participants’ count. However the ‘N’ is already used in the Equation (7) to denote positions’ count. It is better to use another symbol in the Table 1.
  12. Although there is no statistical accuracy in two in-situ porcine study, it is better to give out some results such as the task-completion time, the calibration time, etc. Is the calibration time less than 5 minutes set by author in line 267?

Author Response

Please see the attached Word document.

Round 2

Reviewer 1 Report

Given the recent changes that the authors have made to the manuscript, most of my concerns are addressed/acknowledged. The paper can be accepted in the current form. 

Author Response

Thank you very much for your time and detailed critique throughout the review process.